# Preliminary Design of a Space Habitat Thermally Controlled Using Phase Change Materials

**A. Borshchak Kachalov, P. Salgado Sánchez \*, U. Martínez and J. M. Ezquerro**

E-USOC, Center for Computational Simulation, Escuela Técnica Superior de Ingeniería Aeronaútica y del Espacio, Universidad Politécnica de Madrid, 28040 Madrid, Spain

\* Correspondence: pablo.salgado@upm.es

**Abstract:** We explore the preliminary design of a space habitat thermally controlled using phase change materials (PCMs). The PCM is used to maintain a suitable, habitable temperature inside the habitat by isolating it from the external solar radiation. The system is studied numerically considering only diffusive heat transport (conduction), a scenario with practical application to microgravity or reduced gravity environments. The system dynamics are explored for a wide range of governing parameters, including the length of the PCM cell $L$, the thermo-optical properties—absorptivity $\alpha$ and emissivity $\varepsilon$—at the external boundary of the habitat wall exposed to solar radiation, the eclipse (illumination) fraction $\tau_e$ ($\tau_i$) of the solar cycle, and the PCM used. We find that the thermo-optical properties at the external radiated boundary, characterized by the absorptivity–emissivity ratio ($\alpha/\varepsilon$), play a key role in the system response and largely define the optimal design of the habitat. This optimum balances the heat absorbed and released by the PCM during repeated illumination and eclipse cycles.

**Keywords:** phase change materials; thermal control; microgravity

## 1. Introduction

Motivated in no small part by the emerging environmental awareness of modern societies, studies and applications on the use of phase change materials (PCMs) as thermal control and energy storage devices have notably increased in recent years [1].

PCMs are characterized by the large amount of energy required for the phase transition (latent heat) and the nearly isothermal nature of the phase change. These characteristics can be used in advance for passive temperature control: PCMs can be incorporated to any system so that they absorb energy in the form of latent heat when the system heats up and release it back when it cools down, increasing thermal inertia and maintaining the system temperature around the phase change point: typically, the melting temperature $T_M$. The use of PCMs, therefore, opens the possibility of new passive thermal control devices and energy storage applications. Examples of current relevance can be found in different technological fields ranging from manufacturing [2], food storage [3] and net-zero energy buildings [4,5] to renewable energies [6], electronics [7,8] and space exploration [9,10].

Nowadays, a wide offer of organic and inorganic PCMs is available in the market, with particular (combination of) properties that make them attractive for different applications [11]. Organic PCMs, typified by fatty acids and alkanes, out-stand for their chemical stability and compatibility (non-corrosiveness) with most materials, while displaying a low value of thermal conductivity and a moderate density variation between solid and liquid phases [12]. These latter characteristics may compromise their performance as passive thermal control elements and are related, respectively, with long melting/solidification cycles and with volume changes during the phase change process [13].

In particular, the problem associated with low thermal conductivity and the characteristic low heat transfer rate of organic PCMs has seized a lot of attention. During the last

decades, authors have proposed strategies to palliate it by promoting either conductive or convective transport [14,15]. A straightforward idea results from including high-conductive materials with different shapes (fins, foams, meshes, structures, capsules, etc.) within the PCM device; see Refs. [16,17] and the references therein. Despite the overall efficacy of this solution, it may not be convenient for some applications, as part of the PCM volume is replaced by other (conductive) material that does not undergo the phase transition, and the energy storage capacity of the thermal control device is reduced. Motivated by the same principle, other authors suggested to include dispersed metallic nanoparticles, constructing the so-called Nano-enhanced PCMs (NePCMs) [18,19]. In this case, the heat transport enhancement is relatively small, but the mass and volume of the PCM device is not substantially compromised either. Agglomeration and sedimentation, however, are potential issues associated with NePCMs [20].

Another line of research has looked at heat transport improvements supported by convection within the liquid phase of the PCM. On Earth, natural convection can be exploited to significantly increase (reduce) the heat transfer rate (phase change time) during melting; see Ref. [21] for a detailed review. In microgravity, on the other hand, the thermocapillary effect stands out as the only simple and passive alternative based on convective transport [13]. If the PCM design incorporates a free surface, the temperature gradient inherent to its operation induces variations in surface tension that drive convective flow and enhance the heat transfer rate of the PCM device [22–26]. Furthermore, the presence of a gas layer helps alleviate the problems associated with thermal expansion during melting/solidification. Besides exploiting convective flow, further enhancement can be achieved by using appropriate container geometries on ground [27,28] and in reduced gravity environments [29–31].

From a PCM application perspective, the most extensive literature is found on thermal control in buildings [4,32,33], where the basic requirement refers to maintaining a suitable, habitable temperature in their interior. Nowadays, buildings can achieve this by using energy sources that actively control their internal environment; these are known as *active* buildings. In this context, the concept of *net-zero energy buildings* (NZEBs) is of current relevance and refers to buildings that only use electricity to supplement on-site renewable energy, without adding fuel sources: the annual balance between the energy imported from the electrical grid and that exported to it is zero [34]; including an adequate thermal control can help achieve this goal [35].

Historically, however, buildings had been projected to passively maintain an adequate (habitable) interior temperature. From a design perspective [33], this concept of *passive* building relies on balancing the principal sinks and sources of power affecting the building thermal environment: heat fluxes at its walls and boundaries (solar radiation, radiative emissions, convective cooling, etc.) and its internal energy dissipation. Note the cyclic nature of the environment that a building is exposed to, with different characteristic times ranging from minutes (e.g., fast variations in the internal dissipation) to several hours, days or months (e.g., day/night and seasonal variations on the solar flux). Representative examples of passive thermal control elements in buildings are trombe walls, wall boards, building blocks or ceiling boards, among others.

With their defining energy storage capacity, PCMs can also be a suitable option to passively fulfill the thermal requirements of buildings: they can be incorporated into buildings to absorb/release energy in form of latent heat and, in a passive manner, compensate for potential imbalances in the energy budget [36]. This passive control has been analyzed experimentally in different works: with a PCM wall inside the room [37], using different PCMs with values of $T_M$ in the range of 28–30 °C (SP29) and 17–19 °C (RT18) in accord with the mean seasonal temperature during the year [38], or positioning the PCM in the interior edges for different locations in Europe [39]. In addition, parametric analyses and optimization were performed for the design of an active thermal control system using PCMs for a NZEB; see Ref. [40]. The PCM absorbs heat during the day to maintain comfortable temperature conditions, and the active regeneration of the PCM consists in cooling

the PCM during the night. The reader is referred to Refs. [4,36,41] for detailed reviews of PCM-based thermal control in residential buildings.

PCMs have been used as thermal control devices in space missions too [42]. The work of Arun et al. [9], for instance, analyzed the performance of PCMs in spacecraft transportation, in particular, to maintain the temperature of the (so-called) *Spacecraft Transportation Container* within a given temperature envelope. Other relevant examples of PCM use for thermal control in this field are onboard power generation, where thermal energy stored as latent heat is converted to electrical power, electronic components having cyclic operation conditions and enhancement of thermal systems. As an example, Guan et al. [43] presented the thermal control of a star tracker baffle (aluminum body) using additive manufacturing and n-tetradecane as PCM. Desai et al. [44] investigated a PCM-based thermal control module for satellite components, considering only conductive heat transfer. The operating cycle of the component was assumed to be 100 min:10 min with power generation, and 90 min where it remained powered off. A configuration with fins was studied and optimized numerically.

In this work, we explore the preliminary design of a passive PCM-based thermal system to control the interior temperature of a space habitat. The PCM is integrated in the habitat wall between its exterior and interior boundaries, with the objective of maintaining the interior temperature within an adequate envelope, while the external boundary is subjected to a cyclic, time-dependent solar radiation flux. To the best of our knowledge, no similar studies have been found in the literature. The manuscript is structured as follows. First, the mathematical model is described in Section 2. In Section 3, the influence of the different parameters of the problem is presented and discussed in detail for the alkane n-octadecane, due to its relevance for space exploration [45], and extended to other PCMs of the same family and two types of solar radiation profiles. Conclusions and lines for future work are offered in Section 4.

## 2. Mathematical Formulation

We start introducing the conceptual design of the PCM-based passive thermal control for the space habitat. The habitat consists of a semi-spherical wall of radius $R$, resembling the structure of an igloo that is inflated, as it is pressurized; this type of semi-spherical construction was recently proposed for 3D-printed habitats on Mars [46]. The PCM is placed within the habitat wall—between its exterior and interior boundaries—and acts isolating the internal environment of the habitat from the cyclic variation of the external solar flux. The wall, of thickness $L$ ($\ll$R), displays absorptivity and emissivity values—$\alpha$ and $\varepsilon$, respectively—on its exterior boundary and is composed of multiple (modular) cells of PCM. This concept is sketched in Figure 1. Note that the selection of $\alpha$ and $\varepsilon$ is done by design, using an appropriate painting on the external habitat boundary.

The mathematical model of such PCM-based system, presented below, considers the following simplifications:

1.  The analysis of multiple coupled PCM modules is reduced to the study of a single PCM cell.
2.  Since the habitat would operate in reduced gravity and $L \ll R$, convective flows in (the liquid phase of) the PCM are neglected, and only the conductive transport of heat is considered through the cell.
3.  As a result of these two simplifications, the dynamics of the system are reduced to one dimension [24].
4.  The external and internal boundaries of the wall holding the PCM are considered to be perfectly conducting, i.e., only the PCM undergoing the phase change is analyzed.
5.  The habitat interior temperature matches the PCM temperature at $x = L$ and thus, this boundary is adiabatic for the analysis.
6.  The external boundary is subjected to cycles of illumination and eclipse (denoted by '*i*' and '*e*', respectively) with a period $\mathcal{T}$ so that

$$\mathcal{T}_i + \mathcal{T}_e = \mathcal{T}, \tag{1}$$

or, in dimensionless variables:

$$\tau_i + \tau_e = 1, \tag{2}$$

where $\tau_i$, $\tau_e$ represent the illumination and eclipse fractions, respectively.

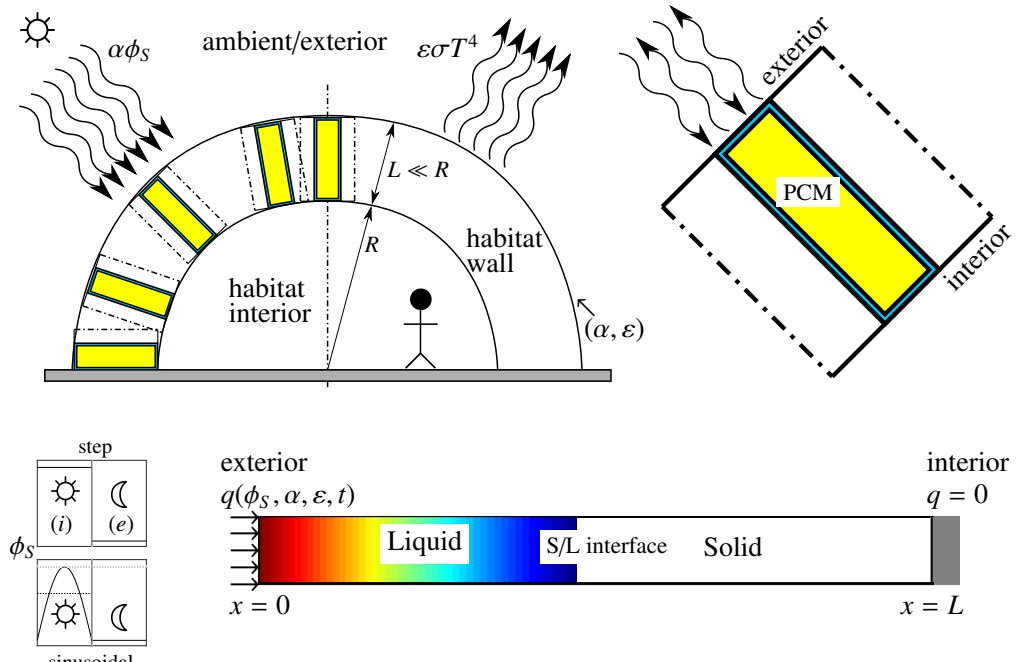

**Figure 1.** Sketch of habitat concept and the associated 1-dimensional model. The main design parameters are the PCM length $L$, the absorptivity $\alpha$ and emissivity $\varepsilon$ at the external radiated boundary, and the selected PCM. The solar cycle is modeled using $\phi_S$ and either a step or a sinusoidal function that accounts for periods of illumination and eclipse. The colormap illustrates a typical temperature field in the liquid phase of the PCM.

When illuminated, the external boundary of the habitat is subjected to solar radiation. To account for periods of illumination and eclipse, we consider two different models in the present analysis. The first model uses the step function $g_1$ that changes from 1 to 0:

$$g_1 = \begin{cases} 1, & t \in [n\mathcal{T}, n\mathcal{T} + \mathcal{T}_i); \\ 0, & t \in [n\mathcal{T} + \mathcal{T}_i, (n+1)\mathcal{T}); \end{cases} \tag{3}$$

where $n = 0, 1, \ldots$ refers to the $(n+1)$-th solar cycle. A more realistic profile is also modeled using the function $g_2$ with $\tau_n = (t - n\mathcal{T})/\mathcal{T}_i$:

$$g_2 = \begin{cases} \sin \pi \tau_n, & t \in [n\mathcal{T}, n\mathcal{T} + \mathcal{T}_i); \\ 0, & t \in [n\mathcal{T} + \mathcal{T}_i, (n+1)\mathcal{T}); \end{cases} \tag{4}$$

which includes a sinusoidal variation during illumination. At the external, radiated boundary $x = 0$; therefore, the absorbed heat flux can be written as

$$q_+ = \alpha \phi_S g_i, \tag{5}$$

where $\alpha$ is the solar absorptivity, $\phi_S = 1368 \ \text{W/m}^2$ is a characteristic value of the solar radiation flux, taken to be that experienced by spacecraft on (low) Earth orbits, and $i = 1, 2$.

Because of its temperature, the external boundary of the habitat radiates heat too. One can model this radiative emission using the Stefan–Boltzmann law:

$$q_- = \varepsilon \sigma T^4, \tag{6}$$

where $\varepsilon$ refers to the (infra-red) emissivity and $\sigma = 5.67051 \times 10^{-8}$ W/(m$^2$ K$^4$) [47,48] is the Stefan–Boltzmann constant. The boundary condition at $x = 0$, therefore, accounts for the (time-dependent) net flux (per unit area):

$$q(t) = \alpha\, \phi_S\, g_i - \varepsilon\, \sigma\, T^4. \tag{7}$$

This boundary condition is indicated in the sketch of Figure 1.

Considering now the steady equilibrium, where fluxes at the exterior boundary are balanced, one can estimate the maximum temperature $T_q$ that can be reached at the habitat external wall ($x = 0$) during illumination:

$$T_q = \left( \frac{\alpha}{\varepsilon}\, \frac{\phi_S}{\sigma} \right)^{1/4}, \tag{8}$$

which only depends on the absorptivity–emissivity ratio $(\alpha/\varepsilon)$. Note that $T_q$ should be greater than the melting temperature $T_M$ of the PCM to drive the phase change, defining a lower bound for the ratio $(\alpha/\varepsilon)$:

$$\left( \frac{\alpha}{\varepsilon} \right)_{\min} = \frac{\sigma}{\phi_S} T_M^4. \tag{9}$$

The value of $T_M$ depends on the selected PCM. Following previous studies, we analyze here various materials from the family of alkanes, as their moderate values of $T_M$ and chemical stability make them attractive for space applications [42]. For its current relevance in the field, n-octadecane is included in the analysis together with n-hexadecane and n-heptadecane, which are selected because their melting temperatures of 18 °C and 21 °C are more suitable for habitable conditions. The thermo-physical properties of these PCMs are indicated in Table 1. Note that the same value of $T_M$ is considered for both melting and solidification processes [49].

**Table 1.** Thermo-physical properties of n-octadecane (nC18), n-heptadecane (nC17) and n-hexadecane (nC16); reproduced from Refs. [12,24]. The subscripts ′*l*′, ′*s*′ refer to the liquid and solid phases of the PCM, respectively. The value $(\alpha/\varepsilon)_{\min}$, as derived in Section 2, is included for its relevance in the analysis.

| Thermo-Physical Property | nC18 | nC17 | nC16 |
|---|---|---|---|
| Melting temperature, $T_M$ (°C) | 28.0 | 21.4 | 18.0 |
| Liquid density, $\rho_l$ (kg m$^{-3}$) | 780 | 772 | 765 |
| Solid density, $\rho_s$ (kg m$^{-3}$) | 865 | 772 | 765 |
| Latent heat, $c_L$ (kJ kg$^{-1}$) | 243 | 165 | 237 |
| Liquid specific heat capacity, $c_{pl}$ (J kg$^{-1}$ K$^{-1}$) | 2196 | 2300 | 2220 |
| Solid specific heat capacity, $c_{ps}$ (J kg$^{-1}$ K$^{-1}$) | 1934 | 1840 | 1950 |
| Liquid thermal conductivity, $k_l$ (W m$^{-1}$ K$^{-1}$) | 0.148 | 0.146 | 0.145 |
| Solid thermal conductivity, $k_s$ (W m$^{-1}$ K$^{-1}$) | 0.358 | 0.200 | 0.330 |
| $(\alpha/\varepsilon)_{\min}$ [–] | 0.341 | 0.312 | 0.298 |

The conservation of energy, describing the (one-dimensional) dynamics of the PCM cell, includes the contributions of sensible heat (temperature variation) and latent heat (phase change) [50]:

$$\rho c_p \frac{\partial T}{\partial t} + \rho c_L \frac{\partial f}{\partial t} = \frac{\partial}{\partial x}\left(k\frac{\partial T}{\partial x}\right), \tag{10}$$

where $\rho$, $c_p$ and $k$ refer to density, heat capacity at constant pressure and thermal conductivity, respectively, $c_L$ is the latent heat of the PCM, and $f$ refers to the local liquid fraction, characterizing the quantity of absorbed (released) heat during melting (solidification) via the product $\rho\, c_L\, f$.

We use a smoothed step function to model the dependence of $f$ on the local temperature $T$ as follows:

$$f(T) = \begin{cases} 0, & T - T_M < -\delta_T/2\,; \\ \frac{1}{2} + \frac{T-T_M}{2\delta_T} + \frac{1}{2\pi}\sin\left(\frac{\pi(T-T_M)}{\delta_T}\right), & |T - T_M| \leq \delta_T/2\,; \\ 1, & T - T_M > \delta_T/2\,. \end{cases} \tag{11}$$

Note that $f$ transitions from 0 (solid) to 1 (liquid) across a temperature interval $\delta_T$ centered at $T_M$. The interval $\delta_T$ is known as the *mushy region*, where the solid and liquid phases of the PCM can coexist [51]. For n-octadecane, a value of $\delta_T = 1$ K is chosen following the works [24,25,52]. For the other PCMs, $\delta_T$ shall be adjusted to allow for comparison [53]. However, since the main goal of the present work is to analyze the design concept of the space habitat, rather than compare the performance of each PCM, a fixed value of 1 K is also used. The thermo-physical properties of the solid and liquid PCM phases ($\rho$, $c_p$, $k$) are expressed using $f$ by writing ($\rho$, $c_p$, $k$) = ($\rho_s$, $c_{ps}$, $k_s$) + ($\rho_l - \rho_s$, $c_{pl} - c_{ps}$, $k_l - k_s$)$f$, where the subscripts 's', 'l' refer to the solid and liquid phases, respectively.

To close the formulation, the aforementioned boundary conditions:

$$-k\frac{\partial T}{\partial x} = \alpha\,\phi_S\,g_i - \varepsilon\,\sigma\,T^4, \quad \text{at} \quad x = 0; \tag{12}$$

$$\frac{\partial T}{\partial x} = 0, \quad \text{at} \quad x = L; \tag{13}$$

are prescribed, together with the initial temperature distribution:

$$T = T_M - \delta_T/2, \quad \text{at} \quad t = 0. \tag{14}$$

*Numerical Model*

The mathematical formulation described above is solved with the finite-element software COMSOL Multiphysics. The time integration is performed using a second-order *backward differentiation formulae* scheme (BDF), initialized with a *backward Euler method*.

The maximum time step $\Delta t$ and element size $\mathcal{S}$ are chosen in accord with convergence tests. Table 2 summarizes the associated results based on the average error $E$ in the liquid fraction $\mathcal{L}$:

$$E = \frac{1}{N\mathcal{T}}\int_0^{N\mathcal{T}} |\mathcal{L} - \mathcal{L}^*|\mathrm{d}t, \tag{15}$$

measured with respect to $\mathcal{L}^*$, obtained with the set of parameters #4 that consider the finest combination. Here, $N$ refers to total the number of periods $\mathcal{T}$ of the solar cycle. We consider $N = 12$ to be representative to identify the PCM behavior at large times ($t \to \infty$), as it will reach a stationary periodic state in time.

**Table 2.** Mesh convergence test for numerical simulations based on the average error $E$ in the liquid fraction $\mathcal{L}$, as defined by Equation (15), and computational cost, measured by the simulation time. Results are shown for n-octadecane, $\varepsilon = 0.66$, $(\alpha/\varepsilon) = 0.48$, $\tau_e = 25\%$, $L = 100$ mm and $g_1$. The selected combination of parameters are marked in bold.

| Parameters | #0 | #1 | #2 | #3 | #4 |
|:---:|:---:|:---:|:---:|:---:|:---:|
| $\Delta t$ (s) | **1/2** | 1/2 | 5 | 1/2 | 1/4 |
| $\mathcal{S}$ (mm) | **1/3** | 2 | 1/3 | 1/6 | 1/6 |
| $E$ (%) | $\mathbf{3.47 \times 10^{-4}}$ | $1.96 \times 10^{-1}$ | $2.29 \times 10^{-2}$ | $3.60 \times 10^{-3}$ | — |
| Sim. time (min) | **137** | 94 | 14 | 213 | 275 |

These results demonstrate that the selected parameters of $\Delta t = 1/2$ s and $\mathcal{S} = 1/3$ mm (labeled with #0 and highlighted in bold) provide a good compromise between reduced error and computational cost. In line with this, the used value of $N = 12$ also represents a good compromise solution between simulation time and required memory.

Furthermore, note that this choice of numerical parameters is also consistent with the work of Salgado Sanchez et al. [52], where an analogous numerical model was validated against microgravity experiments on the melting of n-octadecane, and Salgado Sanchez et al. [24], where the associated phase change dynamics considering only thermal conduction were shown to be one dimensional.

## 3. Results

The main results of this work are presented in this section. First, we estimate different design parameters in Section 3.1, using characteristic values to approximate the energy balance described by Equation (10). Then, in subsequent sections, the effect of each parameter is analyzed and discussed.

### 3.1. Estimates Based on the Energy Balance

Replacing derivatives by increments and using characteristic values for each variable, Equation (10) can be approximated to obtain

$$\frac{\rho c_p \Delta T}{t_c}^{\,0} + \frac{\rho c_L \Delta f}{t_c}^{\,1} \sim \frac{Q}{L}, \tag{16}$$

where the sensible heat is neglected compared to the latent heat, consistent with the large heat of fusion of alkanes. Additionally, note that the characteristic variation of the liquid fraction is order unity, $\Delta f \sim 1$, reflecting that the PCM has completed the phase change.

Considering the boundary condition (12), the characteristic value of $Q$ depends on whether the habitat is illuminated or not:

$$Q_i \sim \alpha \phi_S - \varepsilon \sigma T_M^4, \tag{17a}$$

$$Q_e \sim -\varepsilon \sigma T_M^4, \tag{17b}$$

with the associated characteristic times of each interval $t_c \sim \mathcal{T}_i$, $\mathcal{T}_e$, respectively. Ideally, the integral of these two quantities over the associated $\mathcal{T}_i$, $\mathcal{T}_e$ should match so that the quantity of PCM melted during illumination naturally solidifies during eclipse

$$\left( \alpha \phi_S - \varepsilon \sigma T_M^4 \right) \mathcal{T}_i \simeq \left( \varepsilon \sigma T_M^4 \right) \mathcal{T}_e. \tag{18}$$

Therefore, the optimal selection of the thermo-optical relationship can be expressed as

$$\left( \frac{\alpha}{\varepsilon} \right)_{\text{opt}} = \frac{\sigma}{\phi_S} T_M^4 \, \tau_i^{-1} = \left( \frac{\alpha}{\varepsilon} \right)_{\text{min}} \tau_i^{-1}, \tag{19}$$

which depends on the illuminated fraction of $\mathcal{T}$, defined above as $\tau_i = \mathcal{T}_i/\mathcal{T}$.

Furthermore, the energy evacuated (absorbed) during eclipse (illumination) should be released (stored) by the PCM:

$$\varepsilon\sigma T_M^4 \mathcal{T}_e \simeq \rho c_L L. \tag{20}$$

From a design perspective, one can use these expressions to make an initial estimate of parameters. In particular, considering a given PCM and illumination (or eclipse) fraction $\tau_i$ ($\tau_e$), which is given by the mission, the parameters $(\alpha/\varepsilon)$ and $L$ can be obtained. In Figure 2, the optimal value $(\alpha/\varepsilon)_{\mathrm{opt}}$ defined by Equation (19) is plotted as a function of $\tau_i$ (in %). To determine both thermo-optical properties, note that either $\alpha$ or $\varepsilon$ should be selected a priori. In practice, this is done by selecting an appropriate painting at the external PCM boundary, as noted above.

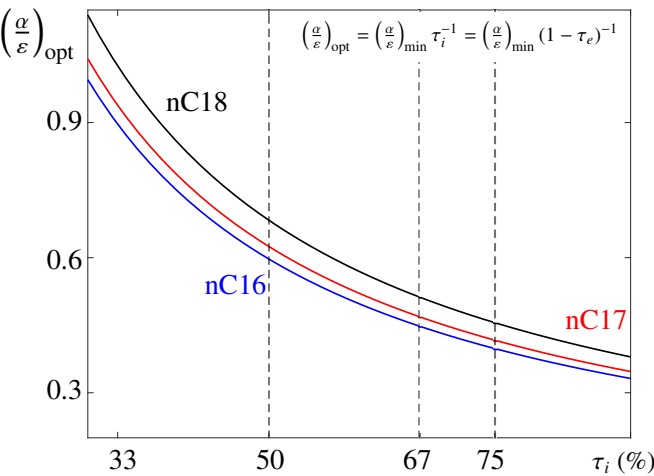

**Figure 2.** Estimate of the optimal thermo-optical relationship, $(\alpha/\varepsilon)_{\mathrm{opt}}$, as a function of $\tau_i$ for n-octadecane (black), n-heptadecane (red), and n-hexadecane (blue). The dashed vertical lines indicate values of $\tau_i$ used in the analysis below.

### 3.2. Effect of the PCM Length and Eclipse Fraction

To analyze the effect of $L$ and $\tau_e$, we select the values $\alpha = \varepsilon = 0.95$, characteristics of a black painting of space use; this type of painting is also known as flat absorber [54]. In addition, the alkane n-octadecane is selected as the PCM, and the cyclic nature of the solar heat flux is modeled using the step function $g_1$ defined in Section 2.

In Figure 3, the time evolution of the temperature $T$ at the external $x = 0$ (red) and internal $x = L$ (blue curves) boundaries of the PCM is illustrated for (a) $L = 22.5$ mm and (b) 100 mm, and an eclipse fraction $\tau_e = 33\%$. As anticipated above, the evolution is shown for 12 periods of the solar radiation cycle, for which a periodic steady state is reached. The value of $T_q \simeq 121\,^{\circ}\mathrm{C}$ is marked with a dashed red line in panel (a).

For $L = 100$ mm [panel (b)], the PCM is able to maintain the internal boundary temperature at $T_M$ for roughly 3 periods. As shown in panel (c), this is associated with the fact that the PCM is not fully melted and the solid/liquid (S/L) front has not reached the position $x = L$, i.e., the liquid fraction $\mathcal{L}$—defined as the fraction of liquid PCM with respect to the total PCM volume—has not reached a value of 100% yet. Note, however, that for this selection of parameters, the PCM finally saturates and reaches a periodic evolution, where $\mathcal{L} = 100\%$ for most parts of the solar cycle and decreases during eclipses up to a value of approximately 80%. In this regime, the internal boundary temperature increases above $T_M$ to finally oscillate around a mean value of 72 °C; see the dashed blue line of panel (b). Once saturated, the temperature difference between the external and internal boundaries of the PCM is associated only with the low thermal conductivity of (liquid) n-octadecane. To help understand the evolution, the video Video S1 is included in Supplementary Materials.

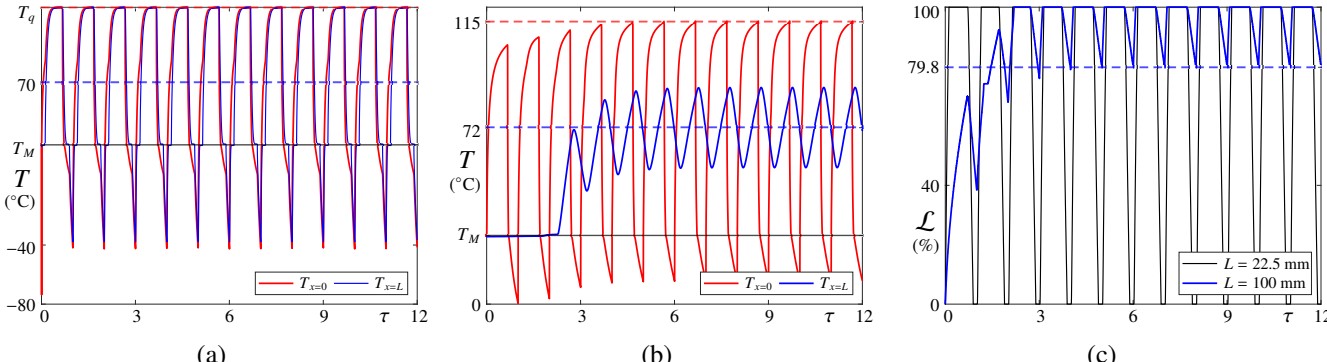

**Figure 3.** (**a**,**b**) Time–evolution of the temperature $T$ at the external $x = 0$ (red) and internal $x = L$ (blue curves) boundaries of the PCM for (**a**) $L = 22.5$ mm and (**b**) 100 mm, and $\tau_e = 33\%$. Results are shown for n-octadecane, $\alpha = \varepsilon = 0.95$, and $g_1$. The horizontal dashed blue line indicates the mean temperature at the stationary state. Panel (**c**) illustrates the associated evolution of the liquid fraction, $\mathcal{L}$.

For $L = 22.5$ mm, this saturation occurs during the first solar cycle and the capability for passive control of the PCM is lost; see the black curve of panel (c). In addition, note that the PCM fully melts and solidifies in each solar cycle, so that $T$ at the internal boundary is maintained at $T_M$ for a short lapse during each period and then oscillates far beyond $T_M$. Again, the temperature difference at both sides of the PCM is due to the low value of $k$. Considering both evolutions, one would seek to find an optimal selection of parameters for which $\mathcal{L}$ oscillates between 0 and 100% without saturating so that the value of $T$ at $x = L$ is ideally maintained at $T_M$, or, in a less restrictive scenario, close to it.

To analyze the effect of the PCM length $L$ in more detail, Figure 4a illustrates the maximum, minimum, and mean steady values of $T$ at the interior PCM boundary as a function of $L$. For $L \lesssim 50$ mm, the PCM melts and solidifies completely during each period. Therefore, the habitat temperature cannot be passively controlled, and it follows the evolution of $T$ at the external boundary with certain delay, as illustrated in Figure 3a. Note that, for $L = 10$ mm, $T$ can eventually reach $T_q$ at some instants of the cycle. By increasing $L$, the PCM saturation is delayed and the variations of $T$ at the interior wall are reduced. Note that the mean stationary temperature is almost invariant. As discussed above, the optimal configuration would be found if temperature oscillations at the interior-adiabatic wall are not present.

Fixing now the cell length at $L = 100$ mm, the mean stationary temperature at the interior wall is analyzed as a function of $\tau_e$, and the results are illustrated in Figure 4b. For small eclipse fractions, the quantity of PCM that solidifies during the eclipse cannot compensate for the quantity of melted PCM during illumination, resulting in a fast saturation of the PCM and in large variations of $T$ and a high mean value. In the opposite case of large $\tau_e$, the melted PCM rapidly solidifies, and the interior boundary cools down far below $T_M$. For this selection of parameters, it is found that the mean temperature remains close to $T_M$ for $\tau_e \simeq 50\%$, resulting in a more effective passive temperature control.

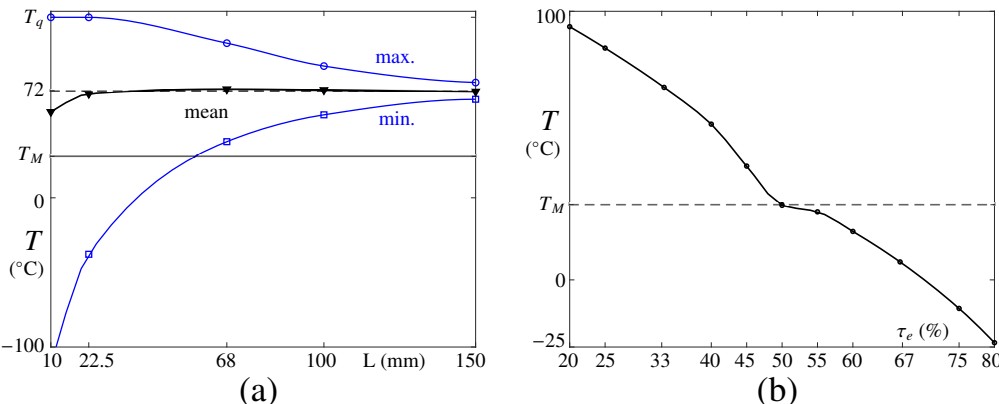

**Figure 4.** (**a**) Maximum, minimum, and mean (labeled) steady values of $T$ at the interior PCM boundary as a function of $L$. (**b**) Mean stationary $T$ at the interior PCM boundary as a function of $\tau_e$ for $L = 100$ mm. Results are shown for n-octadecane.

### 3.3. Effect of the Thermo-Optical Relationship and Selection of Its Optimal Value

Here, a parametric study varying $(\alpha/\varepsilon)$ is conducted. For this purpose, we follow the previous analysis and select n-octadecane as the PCM, use $g_1$ to model the solar cycle, and select $L = 100$ mm. With this choice of parameters, we expect a delay in the PCM saturation. Similarly, we fix $\tau_e = 25\%$ and $\varepsilon = 0.66$—this value is typical of gray paintings [54]. The main goal now is to find the optimal $(\alpha/\varepsilon)$ for which the temperature at the interior wall can be maintained at $T_M$, i.e., it is passively controlled.

Following the idea illustrated in Figure 4, we show in Figure 5 the maximum, minimum and mean steady temperatures at the (a) external and (b) internal boundaries of the PCM as a function of $(\alpha/\varepsilon)$. For reference, the value of $T_q$ (labeled) is included in both panels.

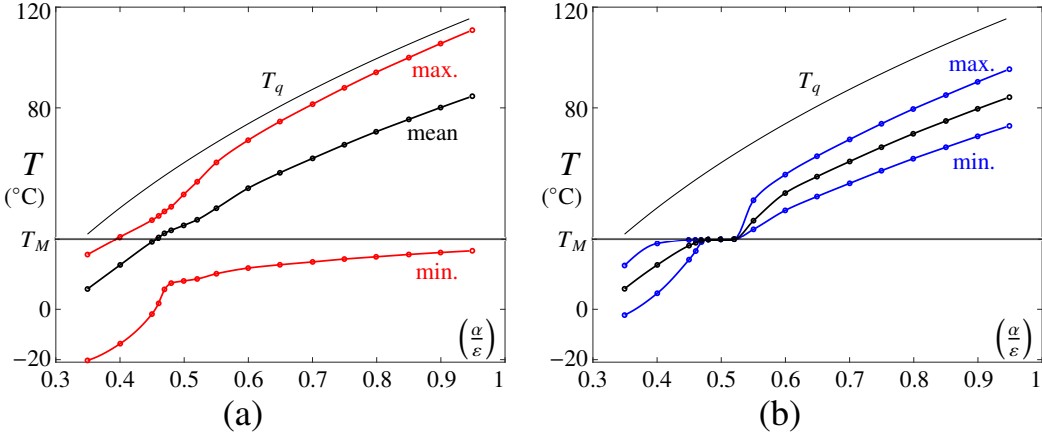

**Figure 5.** Maximum, minimum and mean (labeled) steady $T$ at the (**a**) external and (**b**) internal PCM boundaries as a function of the absorptivity–emissivity ratio, $(\alpha/\varepsilon)$. The value of $T_q$, as derived in Section 2, is included in both panels.

As anticipated earlier, there is a minimum value of $(\alpha/\varepsilon)$ beyond which the PCM does not undergo the phase change. This value can be determined looking at the external wall temperature, where the curve showing its maximum value intersects $T_M$. Numerically, $(\alpha/\varepsilon)_{\min} \simeq 0.4$, which is slightly greater than the predicted value $(\alpha/\varepsilon)_{\min} \simeq 0.34$ obtained using Equation (9). Besides the role of sensible heat that will be discussed below, this difference is attributed to the finite numbers of periods $N$ considered in simulations. Note that if the (maximum) external wall temperature is close to $T_M$, the phase change is driven slowly, and the time for complete melting increases substantially [24,53].

Looking now the internal wall temperature [panel (b)], an interesting interval $(\alpha/\varepsilon) \in (0.45, 0.51)$, where the maximum, minimum and mean values coincide, can be easily identified. To analyze this region in more detail, we present in Figure 6 the associated time evolutions of the temperature (contours) along the complete PCM length for eight periods of the solar radiation cycle and three different values of $(\alpha/\varepsilon) = 0.46, 0.48, 0.5$. The S/L front is superposed to these contours using a solid black line; this line also represents the isotherm $T = T_M$.

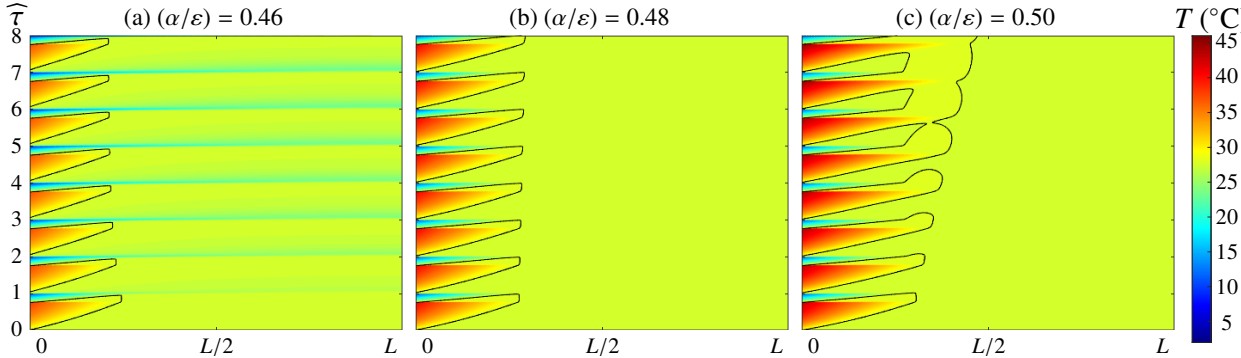

**Figure 6.** Contours showing the time evolution of the temperature field along the PCM length for (**a**) $(\alpha/\varepsilon) = 0.46$, (**b**) 0.48 and (**c**) 0.5 during 8 periods of the solar cycle. The S/L front position is marked using a solid black line. Results are shown for n-octadecane, $\varepsilon = 0.66$, $\tau_e = 25\%$, $L = 100$ mm and $g_1$. Panel (b) corresponds to the optimal value $(\alpha/\varepsilon)_{opt}$.

In panel (a), results are shown for $(\alpha/\varepsilon) = 0.46$. During the illuminated fraction of the solar cycle, the PCM melts, and the S/L front advances toward the interior boundary. Note that part of the absorbed heat is invested in increasing the temperature (sensible heat) of the liquid phase. During eclipse, the PCM solidifies from the external boundary, a process that is completed before the solar cycle ends. This means that the internal boundary temperature is partly affected by the cooling experienced at the external wall, once all the PCM is in solid state.

In panel (c), on the other hand, results are shown for $(\alpha/\varepsilon) = 0.5$ so that the relative importance of the absorbed heat during illumination is slightly greater than in panel (a). The overall dynamics are analogous to that of panel (a), except that the quantity of melted PCM during illumination exceeds the volume of PCM that is solidified during eclipse. Note that the S/L front advances in a secular manner toward $x = L$ (the internal wall of the habitat) after each solar cycle. One would expect, therefore, the PCM saturation at larger times and (small-amplitude) oscillations of the temperature at the internal boundary close to $T_M$.

In panel (b), the results are shown for an intermediate value of $(\alpha/\varepsilon) = 0.48$. Now, the S/L front displays a cyclic evolution where the quantity of PCM melted and solidified during illumination and eclipse, respectively, matches, and the temperature at the internal wall is maintained at $T_M$; therefore, this value of $(\alpha/\varepsilon)$ can be identified as the optimum, $(\alpha/\varepsilon)_{opt}$. Note that the prediction of Equation (19) yields an estimate of the optimal thermo-optical relationship of $(\alpha/\varepsilon)_{opt} \simeq 0.455$, which is approximately 5 % smaller. The fact that the analysis of Section 3.1, where the variation of the PCM temperature (sensible heat) was neglected, underestimates this optimal value, is attributed to the role of this sensible heat in simulations; to melt the same volume of PCM, the quantity of absorbed heat during illumination should be greater in simulations, as part of this heat is invested in increasing the temperature of the PCM. However, note that this error is small, confirming that the estimates discussed in Section 3.1 can be used as starting guesses for further analyses; this also demonstrates that the dynamics are dominated by the latent heat of the PCM. To help understand the evolution, the video Video S2 is included in the Supplementary Materials.

Finally, it is worth noting than the maximum temperature reached (in both boundaries) increases with $(\alpha/\varepsilon)$ and is bounded by $T_q$. Furthermore, note that the PCM acts damping

the temperature oscillations between the exterior and interior walls, increasing the thermal inertia of the system.

### 3.4. Determining the Minimum PCM Length

As described above, Figure 6b shows the optimal configuration where the interior wall temperature remains at $T_M$ and the S/L front moves back and forth over a delimited length of the cell. Therefore, there is a portion of the PCM that *always* remains in a solid state, and it has little to no effect in the system dynamics. In this section, we analyze the minimum cell length $L_{min}$ required to achieve such passive control of the interior wall temperature, for $(\alpha/\varepsilon)_{opt}$. Again, the analysis is presented for n-octadecane and considering the solar cycle associated with $g_1$. We further select an eclipse fraction of $\tau_e = 25\%$.

In Figure 7, the maximum liquid fraction reached, $\mathcal{L}_{max}$, is plotted as a function of the container $L$ for pairs of $(\alpha, \varepsilon)$, whose ratio is optimal. Each set of results can be fitted to $\propto L^{-1}$ so that the value of $L$, for which a maximum liquid fraction of 100% is reached, can be extrapolated. These limiting values of $L = 15, 30, 34$ mm represent the minimum cell length $L_{min}$ (or minimum PCM volume) that would isolate the internal boundary of the PCM from variations of the external fluxes on the habitat, thus achieving an effective temperature control. Note, however, that this lower limit of $L$ is not practical from a design perspective since it leaves no margins for uncertainties.

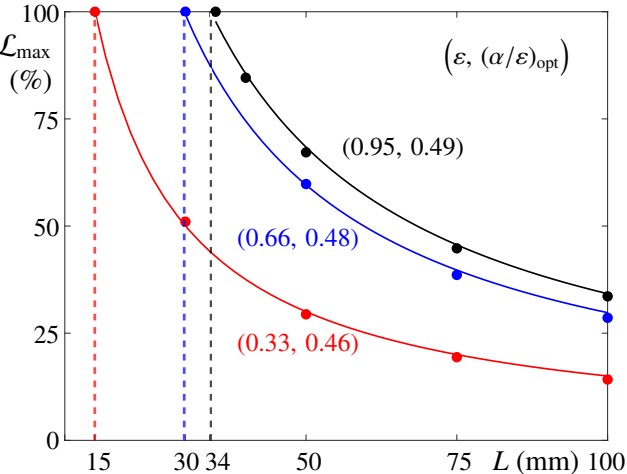

**Figure 7.** Maximum liquid fraction, $\mathcal{L}_{max}$, as a function of $L$ for optimal $(\alpha, \varepsilon)$ combinations. Each set of results fits to $\propto L^{-1}$, allowing to extrapolate the minimum length $L_{min}$ for which a maximum liquid fraction of 100% is reached; these values are indicated with vertical lines.

Again, these numerical results can be compared to the prediction of Equation (20), which allows for an estimate of the minimum PCM length, given by

$$L_{min} = \frac{\varepsilon \sigma T_M^4 \mathcal{T}_e}{\rho c_L}. \tag{21}$$

For n-octadecane, $\tau_e = 25\%$, a solar cycle period equal to the sidereal day $\mathcal{T} \simeq 86{,}400$ s, and three different values of $\varepsilon = 0.33, 0.66, 0.95$, Equation (21) yields $L_{min} \simeq 16, 32, 46$ mm, respectively. Note that these estimates are larger than those predicted numerically, a fact that can be explained again in terms of the sensible heat. In this sense, these estimates are conservative.

### 3.5. PCM Selection and Effect of Sinusoidal Solar Flux

The previous analyses are complemented considering other alkanes: n-hexadecane and n-heptadecane, and using the sinusoidal heat flux profile defined by $g_2$.

The selection of n-hexadecane and n-heptadecane for the present work is motivated by their values of $T_M$, which are closer to typical room temperatures in buildings or the ambient temperature inside the International Space Station (ISS); their properties are summarized in Table 1. Considering this, the use of n-hexadecane and n-heptadecane results in being more attractive than n-octadecane. The analysis presented above, however, remains applicable, and the associated estimates based on the energy equation can be obtained considering their thermo-physical properties. The associated $(\alpha/\varepsilon)_{\mathrm{opt}}$ are illustrated in Figure 2. Note that the optimal thermo-optical relationship decreases with $T_M$, and that one would expect larger differences with respect to simulations for reduced values of $c_L$, as the relevance of the sensible heat is relatively greater.

From the perspective of the more realistic model of the solar cycle $g_2$, the analysis above remains also applicable: one can also estimate the energy balance by averaging the solar flux over one cycle—this approach is typical of preliminary design stages of the thermal control system for space missions. The associated values of $(\alpha/\varepsilon)_{\mathrm{opt}}$ are included in Figure 8. For the sinusoidal profile, the average solar flux during illumination is $\bar{\phi}_S = (2/\pi)\phi_S$, a value that preserves the total quantity of heat absorbed by the PCM compared to the step profile characterized by $\bar{\phi}_S\, g_1$.

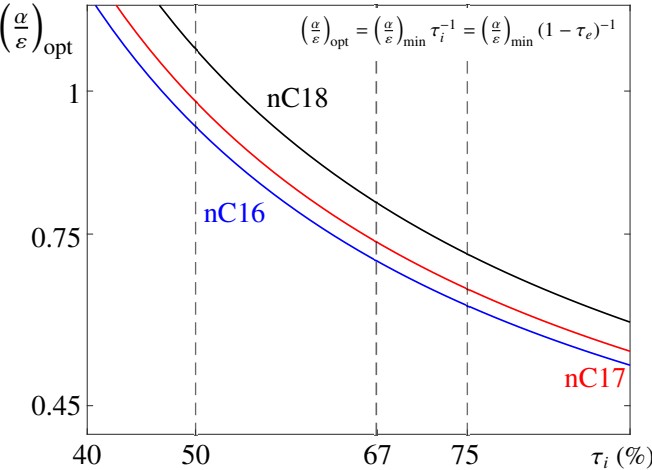

**Figure 8.** Estimate of $(\alpha/\varepsilon)_{\mathrm{opt}}$ as a function of $\tau_i$ for n-octadecane (black), n-heptadecane (red), and n-hexadecane (blue); see Equation (19). Results are calculated for the mean solar flux of $g_2$: $\bar{\phi}_S = (2/\pi)\,\phi_S$.

## 4. Conclusions and Future Work

Nowadays, PCMs are widely used in different applications. In this manuscript, we examined their capability to passively control the thermal environment of a space habitat: the conceptual design integrates the PCM within the habitat walls, reducing the fluctuations of the interior temperature that result from cyclic variations in the solar radiation. To the best of our knowledge, no similar studies were found in literature.

This passive PCM-based system was analyzed using a simplified one-dimensional model, with a time-dependent boundary condition that emulates periods of illumination and eclipse; the internal boundary is considered adiabatic. The main governing parameters were the selected PCM (regarding its properties) and its length $L$, the thermo-optical properties at the external (radiated) wall of the habitat—absorptivity $\alpha$ and emissivity $\varepsilon$—and the solar cycle model $g_i$, including the associated illumination $\tau_i$ and eclipse $\tau_e$ fractions. Due to its relevance to space exploration [42] and previous studies [24,25,52], the organic PCM n-octadecane is selected together with n-heptadecane and n-hexadecane from the same family of materials, whose melting points are more suitable for human living conditions.

The principal goal of the preliminary design was to balance the energy budget of the space habitat by exploiting both the melting and solidification processes of the PCM. The PCM increases the thermal inertia of the habitat wall, allowing to reduce the variations of the internal temperature of the habitat or even suppress them in the optimal scenario.

Neglecting variations in the PCM temperature (sensible heat), initial estimates of the optimal parameters were obtained based on the conservation of energy described by Equation (10). In particular, we obtained the optimal absorptivity–emissivity ratio $(\alpha/\varepsilon)_{opt}$, for which the quantities of PCM melted during illumination and solidified during eclipse match; this optimal value depends on the characteristics of the solar cycle, i.e., the eclipse fraction $\tau_e$.

This basic analysis was extensively complemented with simulations. As predicted by theory, numerical results show that, for a given choice of PCM, geometry, and solar cycle, one can select the optimal combinations of $(\alpha, \varepsilon)$ that effectively maintain the internal temperature of the habitat at $T_M$ in a passive manner. Close to the optimal value $(\alpha/\varepsilon)_{opt}$, small fluctuations of the internal temperature are observed; these are, however, acceptable. The geometry of the PCM can be also optimized, reducing the PCM length $L$. Note that this minimizes the quantity of PCM required to achieve an effective temperature control and, in turn, reduces the associated mass, volume and cost, which are major design drivers in space missions.

For future studies, and with the basic objective of analyzing more realistic scenarios, one could foresee to extend the present model to include the effect of buoyancy and the associated convective transport of heat. In line with this, one could also anticipate interesting results when considering the additional transport supported by thermocapillary convection. We highlight the growing interest of the thermocapillary-driven melting of PCMs, where the thermocapillary effect has been proved as an effective alternative to palliate the problem of low thermal conductivity of organic PCMs in microgravity. In these two scenarios, the analysis can be extremely complicated, given the increasing number of design parameters and the more complex physics involved. For instance, considering the type of semi-spherical habitat studied in this work and the effect of natural convection, each PCM cell would experience a different orientation with respect to the gravity field. These analyses will be undertaken elsewhere.

**Supplementary Materials:** The following supporting information can be downloaded at: https://www.mdpi.com/article/10.3390/thermo3020014/s1.

**Author Contributions:** Conceptualization, A.B.K. and P.S.S.; writing—original draft preparation, all authors; supervision, U.M., P.S.S. and J.M.E.; project administration and funding acquisition, J.M.E. All authors have read and agreed to the published version of the manuscript.

**Funding:** This work was supported by the Ministerio de Ciencia e Innovación under Project No. PID2020-115086GB-C31, and by the Spanish User Support and Operations Centre (E-USOC), Center for Computational Simulation (CCS).

**Data Availability Statement:** Data are contained within the article or Supplementary Material.

**Acknowledgments:** We thank Jeff Porter for helpful discussions and the English revision of the manuscript.

**Conflicts of Interest:** The authors declare no conflict of interest.

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
