# Peer review of "Preliminary Design of a Space Habitat Thermally Controlled Using Phase Change Materials"

_2673-7264, doi:10.3390/thermo3020014_

Round 1

Reviewer 1 Report

The authors investigated a passive PCM-based thermal system to control and maintain the interior temperature of a space habitat. They found that the incorporation of PCM increases the thermal inertia of the habitat walls, thereby reducing variations in the interior temperature, which is logical. The parametric effect of absorptivity-emissivity ratio and PCM length values was discussed to further improve the paper. This article is interesting for this journal, and its content is significant. However, some issues still need to be addressed. With some revisions, the manuscript can be published.

1.  The text appears to be grammatically correct, and there are no obvious spelling or punctuation errors. However, the use of moon and sun symbols is not appropriate for a scientific article.

2. The literature reviews cited in the manuscript lack sufficient statistical and quantitative results that prove the importance of the methods used in this work.

3. The choice of ∆t = 0.5 s and the maximum mesh element size of 1/3 mm needs to be justified with an independence analysis.

4. To better appreciate the model simulated and implemented using COMSOL and to validate the accuracy of the results, it is essential to evaluate them with experimental results or data from the literature.

5. Please add legends to the figure and separate the panels from the figure to improve the visualization of the curves.

6. The authors did not discuss the topic in depth in the discussion section and did not provide their own thinking, reliable solutions, and prospects for the bottleneck of the existing technology.

7. The level of English is adequate, but the text still needs to be revised by a native speaker.

Reviewer 2 Report

The presented paper has some missing information such as:

-                  Defined characteristics of the container/cell material of the PCM integrated in the walls;

-                  Melting temperature for n-octadecane is 27.5 C not 28 regarding to NIST (phase change solid-liquid) (https://webbook.nist.gov/cgi/cbook.cgi?ID=593-45-3)

-                  N-octadecane is used as a solvent and it is flammable and also is classified as health hazard substance(https://pubchem.ncbi.nlm.nih.gov/compound/Octadecane) – I’m not sure if is the best option for use in the proposed application (habitat)

-                  Thermal radiation is not considered in your simulation-please explain

-                  Emissivity value is calculated depending on what properties? (the references are linked to cell material and refractive index in pouch material and MPC content)

-                  Rates of heat propagation/ transfer are not proper explained for the described simulation (transfer in the hole mass of the MPC)

Round 2

Reviewer 1 Report

The quality of the manuscript is now acceptable.